American Society for Microbiology | Microbiology Spectrum

# Integrated Metabolic and Inflammatory Signatures Associated with Severity of, Fatality of, and Recovery from COVID-19

[ORCID] Luiz Gustavo Gardinassi,[a] Carolina do Prado Servian,[a] Gesiane da Silva Lima,[b] Déborah Carolina Carvalho dos Anjos,[a] Antonio Roberto Gomes Junior,[a] Adriana Oliveira Guilarde,[c] Moara Alves Santa Bárbara Borges,[c] Gabriel Franco dos Santos,[b] Brenda Grazielli Nogueira Moraes,[d] João Marcos Maia Silva,[a] Letícia Carrijo Masson,[a] Flávia Pereira de Souza,[a] Rodolfo Rodrigues da Silva,[b] Giovanna Lopes de Araújo,[b] Marcella Ferreira Rodrigues,[b] Lidya Cardozo da Silva,[b] Sueli Meira,[e] Fabiola Souza Fiaccadori,[a] Menira Souza,[a] Pedro Roosevelt Torres Romão,[f] Mônica Spadafora Ferreira,[g] Verônica Coelho,[h,i,j] Andréa Rodrigues Chaves,[b] Rosineide Costa Simas,[b] Boniek Gontijo Vaz,[b] Simone Gonçalves Fonseca[a,j]

[a]Departamento de Biociências e Tecnologia, Instituto de Patologia Tropical e Saúde Pública, Universidade Federal de Goiás, Goiânia, Goiás, Brazil

[b]Laboratório de Cromatografia e Espectrometria de Massas, Instituto de Química, Universidade Federal de Goiás, Goiânia, Goiás, Brazil

[c]Departamento de Medicina Tropical e Dermatologia, Instituto de Patologia Tropical e Saúde Pública, Universidade Federal de Goiás, Goiânia, Goiás, Brazil

[d]Hospital das Clínicas, Faculdade de Medicina, Universidade Federal de Goiás, Goiânia, Goiás, Brazil

[e]Laboratório Profª Margarida Dobler Komma, Instituto de Patologia Tropical e Saúde Pública, Universidade Federal de Goiás, Goiânia, Goiás, Brazil

[f]Laboratório de Imunologia Celular e Molecular, Programa de Pós-Graduação em Ciências da Saúde, Programa de Pós-Graduação em Ciências da Reabilitação, Universidade Federal de Ciências da Saúde de Porto Alegre, Porto Alegre, Rio Grande do Sul, Brazil

[g]Laboratório de Imunogenética, Instituto Butantan, São Paulo, São Paulo, Brazil

[h]Laboratório de Imunologia, Instituto do Coração, Faculdade de Medicina, Universidade de São Paulo, São Paulo, São Paulo, Brazil

[i]Laboratório de Histocompatibilidade e Imunidade Celular, Hospital das Clínicas HCFMUSP, Faculdade de Medicina, Universidade de São Paulo, São Paulo, São Paulo, Brazil

[j]Instituto de Investigação em Imunologia, Instituto Nacional de Ciências e Tecnologia, São Paulo, São Paulo, Brazil

Luiz Gustavo Gardinassi, Carolina do Prado Servian, and Gesiane da Silva Lima contributed equally to this work. Author order was determined based on the type of contribution.

**ABSTRACT** Severe manifestations of coronavirus disease 2019 (COVID-19) and mortality have been associated with physiological alterations that provide insights into the pathogenesis of the disease. Moreover, factors that drive recovery from COVID-19 can be explored to identify correlates of protection. The cellular metabolism represents a potential target to improve survival upon severe disease, but the associations between the metabolism and the inflammatory response during COVID-19 are not well defined. We analyzed blood laboratorial parameters, cytokines, and metabolomes of 150 individuals with mild to severe disease, of which 33 progressed to a fatal outcome. A subset of 20 individuals was followed up after hospital discharge and recovery from acute disease. We used hierarchical community networks to integrate metabolomics profiles with cytokines and markers of inflammation, coagulation, and tissue damage. Infection by severe acute respiratory syndrome coronavirus 2 (SARS-CoV-2) promotes significant alterations in the plasma metabolome, whose activity varies according to disease severity and correlates with oxygen saturation. Differential metabolism underlying death was marked by amino acids and related metabolites, such as glutamate, glutamyl-glutamate, and oxoproline, and lipids, including progesterone, phosphocholine, and lysophosphatidylcholines (lysoPCs). Individuals who recovered from severe disease displayed persistent alterations enriched for metabolism of purines and phosphatidylinositol phosphate and glycolysis. Recovery of mild disease was associated with vitamin E metabolism. Data integration shows that the metabolic response is a hub connecting other biological features during disease and recovery. Infection by SARS-CoV-2 induces concerted activity of metabolic and inflammatory responses that depend on disease severity and collectively predict clinical outcomes of COVID-19.

**IMPORTANCE** COVID-19 is characterized by diverse clinical outcomes that include asymptomatic to mild manifestations or severe disease and death. Infection by SARS-CoV-2 activates inflammatory and metabolic responses that drive protection or

Address correspondence to Luiz Gustavo Gardinassi, luizgardinassi@ufg.br, Rosineide Costa Simas, simas.rc@gmail.com, Boniek Gontijo Vaz, boniek@ufg.br, or Simone Gonçalves Fonseca, sfonseca@ufg.br.

The authors declare no conflict of interest.

pathology. How inflammation and metabolism communicate during COVID-19 is not well defined. We used high-resolution mass spectrometry to investigate small biochemical compounds (<1,500 Da) in plasma of individuals with COVID-19 and controls. Age, sex, and comorbidities have a profound effect on the plasma metabolites of individuals with COVID-19, but we identified significant activity of pathways and metabolites related to amino acids, lipids, nucleotides, and vitamins determined by disease severity, survival outcome, and recovery. Furthermore, we identified metabolites associated with acute-phase proteins and coagulation factors, which collectively identify individuals with severe disease or individuals who died of severe COVID-19. Our study suggests that manipulating specific metabolic pathways can be explored to prevent hyperinflammation, organ dysfunction, and death.

**KEYWORDS** COVID-19, data integration, inflammation, metabolomics

The clinical spectrum of infection by severe acute respiratory syndrome coronavirus 2 (SARS-CoV-2) ranges from an asymptomatic state to various manifestations of coronavirus disease 2019 (COVID-19) and death. Despite the success of vaccination in preventing severe disease and mortality, a better understanding of the molecular mechanisms of COVID-19 pathogenesis is required to develop precise and accessible therapy. Factors associated with severe disease and death include age, sex, and comorbidities that seem to amplify viral replication and inflammatory responses (1, 2).

Inflammation is a critical component of severe COVID-19, and an immunological dysregulation is associated with unfavorable clinical outcomes (3). There is still limited knowledge about factors driving asymptomatic infection or mild disease. At the same time, post-COVID-19 survival and recovery have not been well characterized at the molecular level. Hematological and biochemical parameters as well as cytokines and chemokines, bioactive lipids, and metabolites are markedly modulated during SARS-CoV-2 infection and have been assessed for their diagnostic and prognostic potential (4–9). Specific cytokines, such as interleukin 6 (IL-6) and IL-1$\beta$, have been evaluated as therapeutic targets (10, 11). However, broad suppression of the inflammatory response via corticosteroid and JAK-STAT inhibitors has promoted better improvement than therapies targeting specific cytokine signaling (12, 13). Therefore, the concerted activity of multifactorial processes rather than signaling mediated by single candidate molecules could better explain the complexity of the pathophysiology of COVID-19 and its diverse clinical outcomes.

Cellular activation and function are tightly interwoven with metabolic adaptations, which support energetic demands, provide building blocks for catabolic or anabolic activity, and mediate cell signaling. Perturbations of homeostasis, such as infections and inflammation, can be characterized by metabolic fingerprints, which in turn provide information on molecular mechanisms of pathogenesis (14). High-resolution tandem mass spectrometry coupled with liquid chromatography (LC-MS/MS) is a powerful metabolomics platform to measure small molecules and lipids of <2,000 Da at large scales and has been helpful for dissecting metabolic profiles of blood from individuals with COVID-19 (15–17). However, metabolome-wide associations with inflammatory responses upon severe COVID-19, but especially death and recovery, are not well defined. Here, we performed an untargeted metabolomics study employing data-driven approaches to determine the metabolic profiles of individuals with COVID-19 and understand how they are associated with inflammation under diverse circumstances.

We found a significant influence of confounding factors, such as sex, age, and comorbidities, on the metabolomes of individuals with COVID-19. After controlling for these covariables, severity of disease still dictated the metabolic activity in the plasma, revealing metabolic pathways that also correlate with oxygen saturation. Modulation of glutamate, tryptophan, oxoproline, progesterone, lysophosphatidylcholines (lysoPCs) and related metabolites characterizes fatal disease, and patients who recover from acute COVID-19 display persistent metabolic alterations in the plasma. Importantly, metabolites are integrative hubs linking cytokines, inflammatory factors, and coagulation functions in

multifactorial, multiscale networks that predict the severity of, fatality of, and recovery from COVID-19.

## RESULTS

**Demographic characteristics and inflammatory profiles of individuals with COVID-19.** We recruited 150 individuals who tested positive for SARS-CoV-2 by reverse transcription-quantitative PCR (RT-qPCR) or serology between June 2020 and February 2021. Twenty-seven individuals with negative RT-qPCR and serology for SARS-CoV-2 were enrolled as control donors. Individuals with COVID-19 were stratified into mild, moderate, and severe disease based on clinical parameters, including oxygen saturation and admission to an intensive care unit, in addition to those progressing to a fatal outcome (see Table S1 in the supplemental material). Longitudinal follow-up of a subset of individuals ($n = 20$) allowed paired sampling after recovery from COVID-19 (Table S2). The analysis of clinical profiles considering the individuals' sex revealed a higher proportion of men with fatal COVID-19 (Table S1). The distribution of age was similar between controls and individuals with COVID-19, except for individuals with mild disease (Table S1). Demographic, clinical, and laboratorial data, including the description of comorbidities, symptoms, and treatments, are detailed in Tables S1 and S2.

Hematological characteristics and levels of biochemical laboratory parameters related to organ damage, acute-phase proteins, and coagulation functions were similar to those found in other cohorts (4, 18, 19). For example, individuals with fatal COVID-19 displayed lymphopenia, while individuals with mild disease and recovered individuals exhibited cell counts comparable to those of control donors (Fig. S1A). Neutrophil counts increased in individuals with moderate, severe, and fatal disease compared to control donors, with values in the group with fatal outcomes also differing from those in other groups (Fig. S1A). Levels of C-reactive protein (CRP), creatinine, and ferritin also increased in the plasma of individuals with fatal disease in comparison to almost all other groups (Fig. S1B). Relative abundance of most cytokines was elevated in the plasma of individuals with moderate, severe, and fatal COVID-19, with IL-6 reaching the highest levels among them (Fig. S1C). Compared to control donors, individuals who recovered from acute COVID-19 displayed small but significant alterations in the abundance of plasma cytokines (Fig. S1D).

**The metabolomes of individuals with COVID-19 are influenced by confounding factors.** To investigate the metabolic responses to SARS-CoV-2 infection, we performed an untargeted metabolomics analysis of plasma samples from individuals with COVID-19 and compared them to those from control donors. There were 578 up- and 425 downregulated metabolite features (false discovery rate [FDR] $< 0.0001$) in the plasma samples from individuals with COVID-19 (Fig. 1A). Hierarchical clustering based on significant features revealed two major clusters of individuals with positive or negative SARS-CoV-2 infection, except for a few outliers (Fig. 1B). Mummichog software analysis, which was designed to evaluated untargeted metabolomics data (20), predicted the activity of diverse metabolic pathways, including arachidonic acid metabolism and arginine and proline metabolism, among others that have been observed in independent cohorts of individuals with COVID-19 (Fig. 1C) (21). Among the significant metabolites, infection was associated with increased abundance of plasma carnitine, acetylcarnitine, butyrylcarnitine, methylglutarylcarnitine, octenoylcarnitine, and decenoylcarnitine (Fig. 1D).

The high heterogeneity of metabolomic signatures among individuals with COVID-19 suggests that intrinsic factors such as age or sex could affect levels of circulating metabolites and thus confound the results. To test whether these covariables could be confounding factors, we first performed logistic regressions to identify statistical associations between metabolite features and age, sex, heart disease, hypertension, diabetes, obesity, dyslipidemia, chronic renal disease (CRD), and chronic obstructive pulmonary disease (COPD) in our study cohort. All the tested variables were associated significantly ($P < 0.05$) with many metabolite features (Fig. 1E), while several metabolite features affected by SARS-CoV-2 infection were the same as those associated with covariables (Fig. S2).

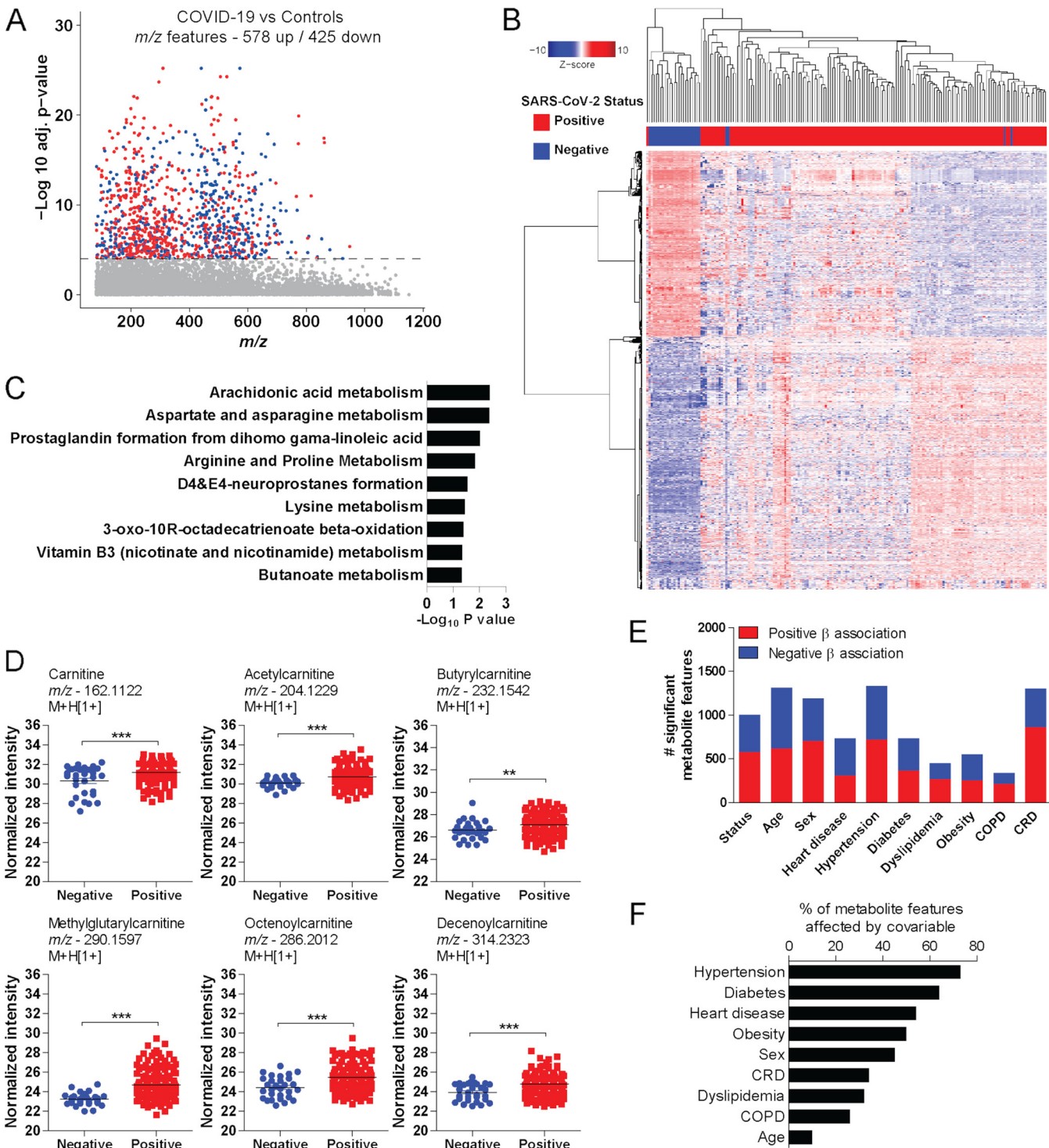

**FIG 1** Plasma metabolomic signatures of SARS-CoV-2 infection. (A) Manhattan plot depicting significant metabolite features in plasma of individuals with COVID-19 compared to control donors. The dashed line indicates an FDR-adjusted *P* value of <0.0001. Upregulated features are in red and downregulated features in blue. (B) Two-way hierarchical clustering based on significant metabolite features. (C) Mummichog pathway analysis of significant features. (D) Differential abundance of carnitine and acylcarnitines according to infection status. (E) Number of significant metabolite features associated with SARS-CoV-2 infection, or age, sex, COPD, CRD, diabetes, dyslipidemia, heart disease, hypertension, and obesity. (F) Percentage of metabolite features whose coefficient of association with infection changes by more than 10% with the inclusion of the covariable in the logistic regression model. Tukey's multiple-comparison test was used in additional statistics. **, *P* < 0.01; ***, *P* < 0.001.

Because of the unbalanced number of individuals with comorbidities in our cohort (e.g., different sample sizes of individuals with diabetes and those with CRD), we used a different approach to understand the impact of covariables as confounding factors. We used logistic regression to calculate the metabolome-wide association with infection, comparing models that included the covariable of interest or not. For that, we calculated the difference in the measurement of association (regression coefficient) resulting from the two models and considered an effect if the difference was 10% or more for all metabolite features. We found that, to some extent, all the covariables influenced the plasma metabolome, where hypertension and diabetes were the major confounding variables (Fig. 1F). Therefore, the metabolomes of individuals with COVID-19 are influenced by factors beyond infection that can affect data interpretation.

**Differential metabolic activity according to COVID-19 severity.** To identify metabolic correlates of disease severity, we compared individuals with mild, moderate, severe, or fatal COVID-19 to control donors, accounting for confounding factors in the linear regression models. The number of significant metabolite features (FDR $< 0.0001$) varied by severity classification, but individuals with fatal COVID-19 displayed the highest numbers of both up- and downregulated features (Fig. 2A). The abundance of carnitine (Fig. S3C) and phenylalanine (Fig. 2B) increased in the plasma of individuals with COVID-19 compared to controls but was similar among categories of severity. In agreement with previous findings (8), the abundance of $N1$-acetylspermidine increased with COVID-19, and it was even higher in the plasma of individuals with fatal outcomes (Fig. 2C). Fatal COVID-19 was also associated with increased abundance of glutamyl-glutamate in the plasma (Fig. 2C). Mummichog pathway analysis predicted the activity of several metabolic pathways according to each clinical stratification (Fig. 2D). As expected, fatal disease was associated with more pathways than the other categories of severity. Butanoate metabolism was significant for all groups of individuals with COVID-19 compared to control donors. Pathways such as arginine and proline metabolism, aspartate and asparagine metabolism, and lysine metabolism were enriched for mild and fatal disease, while prostaglandin formation from dihomo-$\gamma$-linoleic acid was enriched for moderated and fatal disease (Fig. 2D). Because of the significant effect of infection over the plasma metabolome, we then compared only the metabolomes of individuals with COVID-19. For that, we used a limma-moderated $F$ test, which revealed 396 significant metabolite features (FDR $< 0.0001$) (Fig. S3A). Mummichog analysis predicted the activity of most of the same pathways when each group of individuals was compared to control donors (Fig. S3B).

Oxygen saturation (SpO$_2$) is an important indicator of severity and has been used in clinical prediction scores (22, 23). We performed a metabolome-wide association analysis of SpO$_2$ adjusted for confounding factors to identify metabolites predicting this clinical feature. We identified 149 significantly associated metabolite features ($P < 0.001$), of which 44 were negatively associated and 105 were positively associated with SpO$_2$ in individuals with COVID-19 (Fig. 2E). Significant metabolites were predicted to be involved with tryptophan metabolism, histidine metabolism, and arginine and proline metabolism (Fig. 2F). At the metabolite level, an example of negative association includes creatine, while positive associations include aminolevulinate, lysine, and urocanate (Fig. 2G).

**Fatal COVID-19 is characterized by the modulation of lipids and amino acids.** To identify metabolites that are associated with fatal outcomes upon severe disease, we compared the metabolic activity between individuals with fatal disease and those that survived severe COVID-19. To account for the abundance from control donors before comparing the groups of COVID-19 patients, we subtracted the mean log$_2$ intensity values of the control donors from those of COVID-19 samples. The resulting values reflect how much the abundance of metabolites from each individual in both groups changed relative to the mean abundance of control donors. The analysis revealed 158 down- and 393 upregulated metabolite features in the plasma of individuals with fatal disease (Fig. 3A). We detected reduced abundance of lipids such as progesterone, phosphocholine, lysoPC (16:0), and lysoPC(20:4) (Fig. 3B). Significant metabolite features show some heterogeneity in abundance among individuals who survived, which is less apparent in individuals with

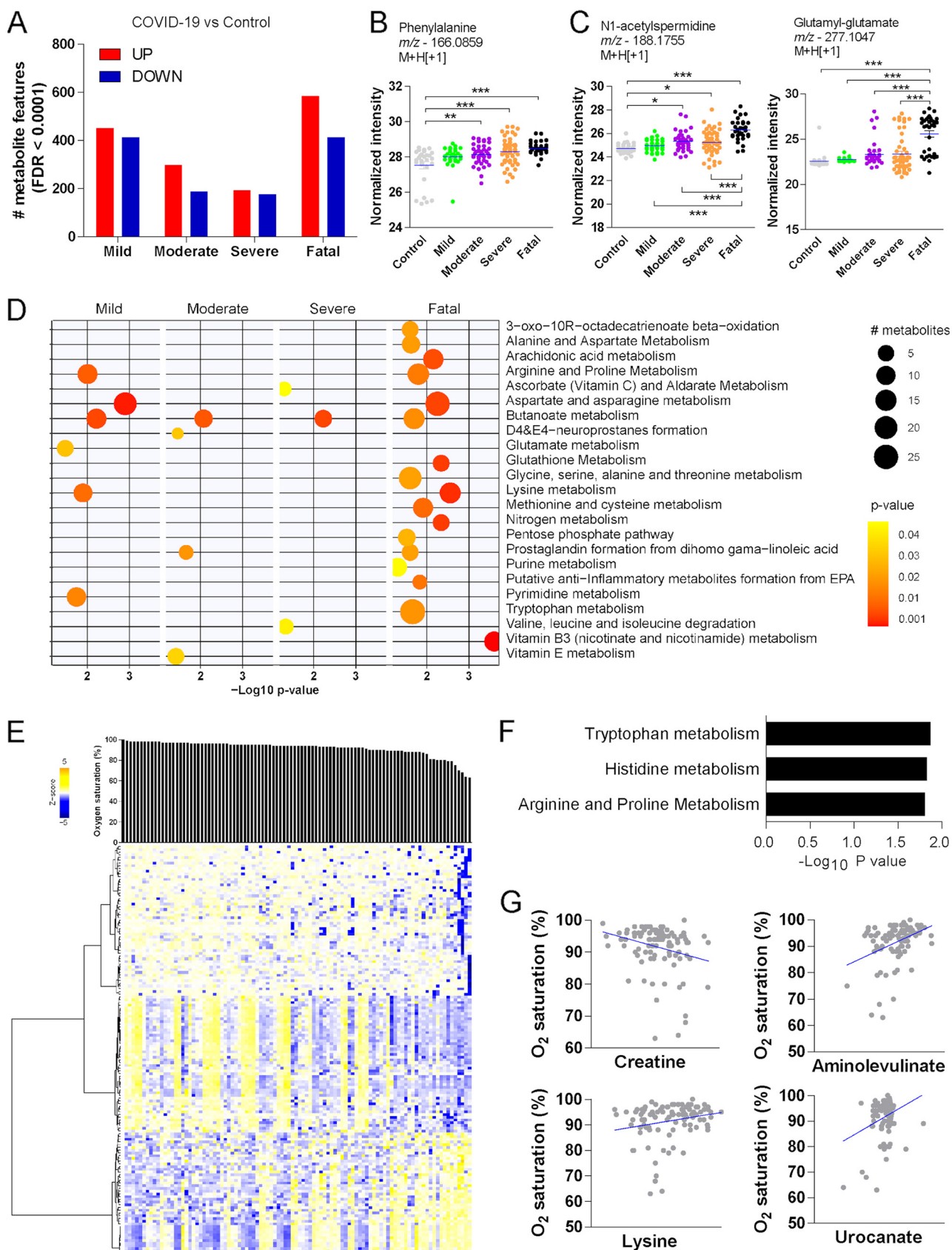

**FIG 2** Metabolomic phenotypes according to severity of COVID-19. (A) Significant metabolite features between individuals with COVID-19 stratified by disease severity compared to control donors. (B) Differential abundance of phenylalanine. (C) Differential abundance of *N*1-acetylspermidine and

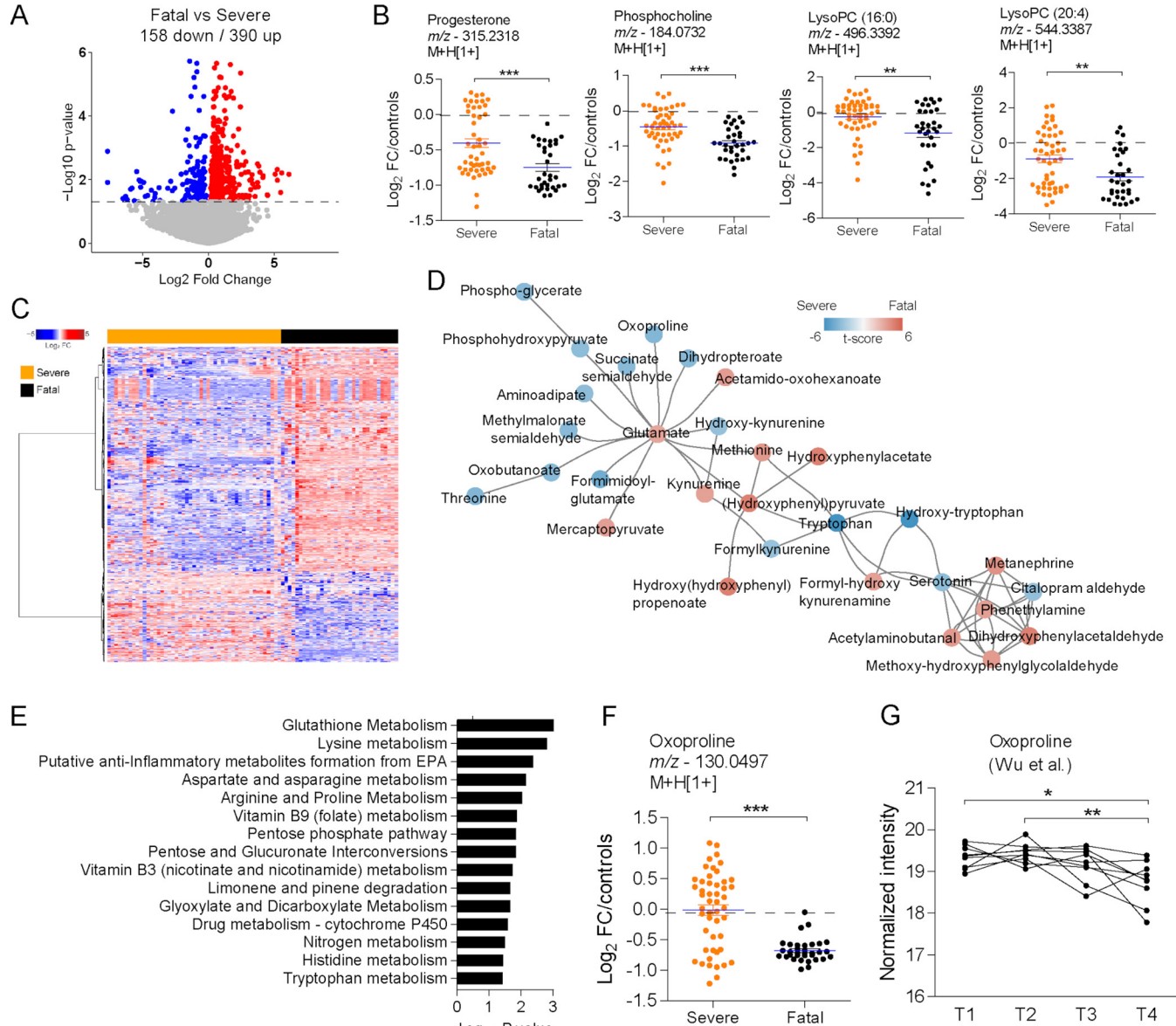

**FIG 3** Metabolic correlates of fatal COVID-19. (A) Volcano plot demonstrating metabolite features differing in the plasma of individuals with severe or fatal COVID-19. The dashed line indicates an FDR of <0.05. (B) Differential abundance of progesterone, phosphocholine, lysoPC(16:0), and lysoPC(20:4) according to survival of severe disease or fatal outcome. (C) One-way hierarchical clustering based on significant metabolite features. (D) Mummichog modular analysis depicting a metabolic network of amino acids and derived metabolites. (E) Mummichog pathway analysis of significant metabolite features. (F) Differential abundance of oxoproline according to survival of severe disease or fatal outcome. (G) Reanalysis of the relative abundance of oxoproline along the course of 4 different time points before death due to COVID-19, retrieved from the study by Wu et al. (16). *, $P < 0.05$; **, $P < 0.01$; ***, $P < 0.001$.

fatal outcome (Fig. 3C). Mummichog analysis predicted a significant metabolic network ($P < 0.05$) of amino acids and derived metabolites (Fig. 3D). Mummichog analysis also predicted activity of pathways such as glutathione metabolism, lysine metabolism, and arginine and proline metabolism (Fig. 3E). Levels of oxoproline (involved in glutathione metabolism) were reduced in the plasma of individuals with fatal COVID-19 (Fig. 3F). Reanalysis of longitudinal data from an independent cohort that progressed to a fatal out-

**FIG 2** Legend (Continued)
glutamyl-glutamate according to disease severity. (D) Pathway enrichment analysis of significant metabolite features. (E) Hierarchical clustering based on significant metabolites associated with oxygen saturation (SpO₂) in individuals with COVID-19. (F) Mummichog pathway analysis of metabolite features associated with SpO₂. (G) Examples of metabolites associated with SpO₂ ($P < 0.05$). Tukey's multiple-comparison test was used in additional statistics. *, $P < 0.05$; **, $P < 0.01$; ***, $P < 0.001$.

come (16) demonstrates a gradual reduction in the abundance of plasma oxoproline with time (Fig. 3G).

**Integrated molecular networks associated with severity and fatality of COVID-19.** Many studies demonstrate metabolites as key components of the host response to infection and vaccines (24–26), suggesting concerted activity of the metabolic and inflammatory responses during COVID-19. We used a hierarchical community network approach (24) to integrate hematological, acute-phase/coagulation, cytokine, and metabolomics data from 53 individuals with COVID-19 (Fig. S4). This method accounts for dimension reduction with meaningful grouping of features and different variance structures of distinct data types, because associations are tested via partial least-square (PLS) regression and significance is assessed by permutation, and therefore, only robust signals reach statistical significance (25). The leading network is composed of 27 nodes and 130 connections (edges at $P < 0.05$; 44 edges at $P < 0.001$) with 1 hematological cluster, 3 biochemical clusters, 2 cytokine clusters, and 21 metabolite clusters (Fig. 4A). Associations between a few nodes are highlighted in purple in Fig. 4A and enlarged in Fig. 4B, which shows that cytokine cluster 2 is composed of tumor necrosis factor alpha (TNF-$\alpha$), IL-2 and IL-4; biochemical cluster 4 is composed of prothrombin activity time (PAT), activated partial thromboplastin time (aPTT), and international normalized ratio (INR); hematological cluster 2 is composed of neutrophils, monocytes, and total leukocytes; and metabolite cluster 11, which connects the different data types, is related to mitochondrial pathways, including electron transport chain, fatty acid oxidation, and tricarboxylic acid (TCA) cycle. The most significant association ($P < 2.2E{-}17$) was between hematological cluster 2 and metabolite cluster 11 (Fig. 4C), supporting our previous findings of increased mitochondrial activity in leukocytes of individuals with COVID-19 (27).

We queried the main network in search of subnetworks that could explain differences in disease severity and fatality of COVID-19. For that, we used the rank-based method GSEA (gene set enrichment analysis) using the network nodes as gene sets and permutation to test the significances. Comparison between individuals with mild or moderate and severe COVID-19 revealed biochemical cluster 2 and several metabolite clusters differentially associated with disease severity (Fig. 4D). Metabolite cluster 19 displayed many upregulated metabolites in plasma of individuals with severe disease (Fig. 4E), which are related to carbohydrate metabolism pathways such as galactose metabolism, amino sugar metabolism, and hexose phosphorylation (Fig. 4F). We also queried the leading network to identify clusters associated with fatal outcomes of COVID-19 (Fig. 4G). Biochemical cluster 2 (Fig. 4H) displayed increased activity in individuals with fatal disease (Fig. 4I). Furthermore, metabolite cluster 21 also exhibited increased activity in individuals with fatal COVID-19 (Fig. 4J) and is related to inflammatory lipid pathways, including linoleate metabolism and arachidonic acid metabolism, among others (Fig. 4K).

**COVID-19 induces persistent metabolic alterations in the blood.** To identify metabolic correlates of recovery from COVID-19, we followed up a subset of 20 individuals with varying clinical severity after recovery of acute disease. Compared to control donors, there were 493 significant metabolite features (FDR $< 0.0001$) in the plasma of recovered individuals (Fig. S5A). Predicted activity was related to pathways such as aspartate and asparagine metabolism, arginine and proline metabolism, and others (Fig. S5B). Of note, levels of adenine and guanine were higher in the plasma of recovered individuals, while the abundance of cytosine and spermidine decreased after recovery of acute COVID-19 (Fig. S5C).

Analysis of paired samples of the same group of individuals demonstrated 143 down- and 309 upregulated metabolite features (Fig. 5A), with significant increases in the abundance of metabolites such as adenine and theobromine after recovery from COVID-19 (Fig. 5B). Mummichog analysis predicted the activity of a few pathways, including phosphatidylinositol phosphate metabolism, glycolysis and gluconeogenesis, and vitamin E metabolism (Fig. 5C). Stratification by disease severity showed many significant features in the plasma of individuals who recovered from severe COVID-19 but not in those who recovered from mild or moderate disease (Fig. 5D). Pathway analysis

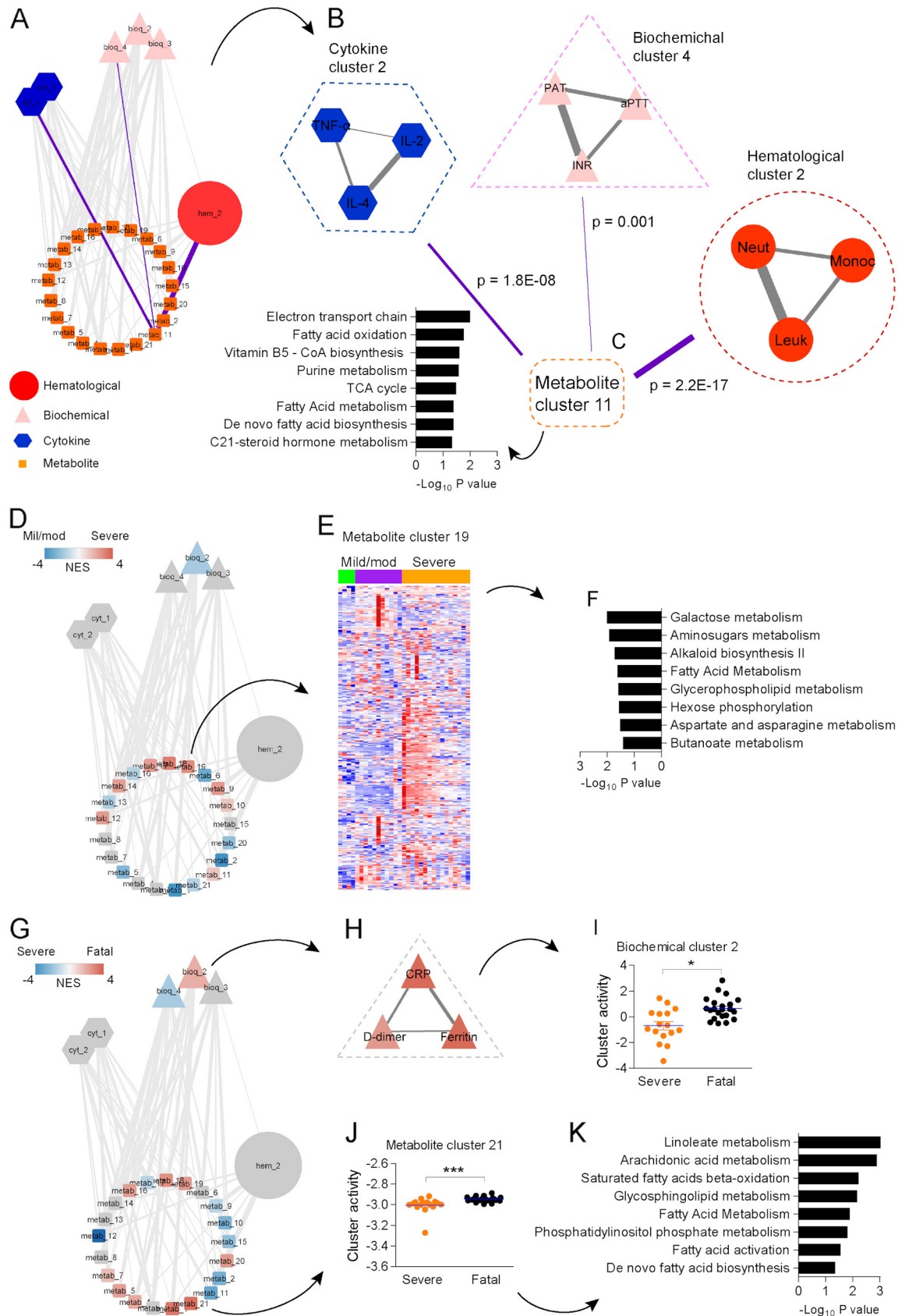

**FIG 4** Integrative hierarchical community network underlying COVID-19. (A) The main network is composed of subnetworks (communities) of hematological data (hematological cluster), laboratory data (biochemical clusters), cytokine data (cytokine clusters)

predicted that activity of vitamin E metabolism is driven by recovery of mild disease (Fig. 5E, green bar), while other pathways such as purine metabolism, phosphatidylinositol phosphate metabolism, and glycolysis and gluconeogenesis are related to recovery of severe disease (Fig. 5E, orange bars). The abundance of the feature annotated as carboxy-tocotrienol increased in the plasma of individuals recovering from mild disease, while the feature annotated as phosphohydroxypyruvate increased in the plasma of individuals recovering from severe disease (Fig. 5F).

We used the hierarchical community network method to integrate hematological, cytokine, and metabolomics data from individuals who recovered from COVID-19. For this, we normalized data by subtracting measurements at time point 1 (disease) from those at time point 2 (recovery). Following, we applied hierarchical clustering followed by PLS regression to find associations, whose significance was tested on 1 million permutations. The network contains 18 nodes with 38 connections (at $P < 0.05$; 10 connections at $P < 0.001$) (Fig. 5G). Highlighted connections are displayed in Fig. 5H, showing that metabolite cluster 12 is enriched by glycerophospholipid metabolism or arachidonic acid metabolism and exhibits a single association with cytokine cluster 1. Cytokine and hematological clusters connect to metabolites, such as metabolite cluster 19, involved in xenobiotic metabolism or prostaglandin formation (Fig. 5H). Comparing the activities of cytokine and hematological clusters between individuals recovering from mild/moderate and severe disease revealed no differences (Fig. 5I). However, individuals recovering from severe COVID-19 exhibited reduced activity of metabolite cluster 1, which is related to steroid hormone biosynthesis, and metabolite cluster 4, which is related to phytanic acid oxidation (Fig. S6). In contrast, they displayed increased activity of metabolite cluster 7, which is related to drug metabolism, as well as metabolite cluster 13, which is associated with linoleate metabolism (Fig. S6).

## DISCUSSION

We employed an untargeted metabolomics approach to assess the molecular phenotypes in the plasma of individuals with mild to fatal COVID-19, and a subset that recovered from acute disease. Furthermore, we integrated biological factors that mediate and are affected by inflammation with the metabolic responses of individuals with COVID-19. This is important, because the cellular metabolism represents a potential target to modulate the inflammatory response, and therefore, there is great interest in discovering how metabolites impact COVID-19 (9, 15–17, 21, 28–33).

Our study demonstrates that infection by SARS-CoV-2 induces important adaptations in the metabolism that can be detected in the plasma. These alterations reflect the activity of cells in the upper and lower respiratory tracts but can also originate from other organs in response to inflammatory cues. Many of the detected metabolites and pathways are involved in immune and inflammatory responses. For instance, arachidonic acid metabolism and generation of eicosanoids are involved in diverse mechanisms of innate and adaptive immunity and have been explored during COVID-19 (34–36). In our study, arachidonic acid metabolism was mostly linked to death, suggesting eicosanoids as amplifiers of the inflammatory response. Furthermore, increased abundance of carnitine and acylcarnitines in plasma during COVID-19 suggests disruptions in mitochondrial activity and fatty acid oxidation and can be related to poorer

**FIG 4** Legend (Continued)
and metabolomics data (metabolite clusters), which are represented by nodes. Significant associations obtained via PLS regression and permutation test are represented by edges between nodes. (B) Amplified visualization of nodes linked by purple highlighted edges on the main network. (C) The most significant association in the network between metabolite cluster 11 and hematological cluster 2. (D) Communities related to the severity of COVID-19, colored by normalized enrichment score (NES) in a comparison between individuals with mild/moderate and severe COVID-19. Red indicates higher association with severe disease. (E) Heat map depicting the abundance of metabolite features composing metabolite cluster 19 in individuals with mild/moderate and severe disease. (F) Mummichog pathway analysis of metabolite cluster 19. (G) Communities related to fatality of COVID-19, colored by NES in a comparison between individuals with severe and fatal disease. Red indicates a higher association with death. (H) Amplified visualization of biochemical cluster 2. (I) Differential cluster activity of biochemical cluster 2 between individuals with severe and fatal disease. (J) Differential cluster activity of metabolite cluster 21 between individuals with severe disease and fatal disease. (K) Mummichog pathway analysis of metabolite cluster 21. *, $P < 0.05$; ***, $P < 0.001$.

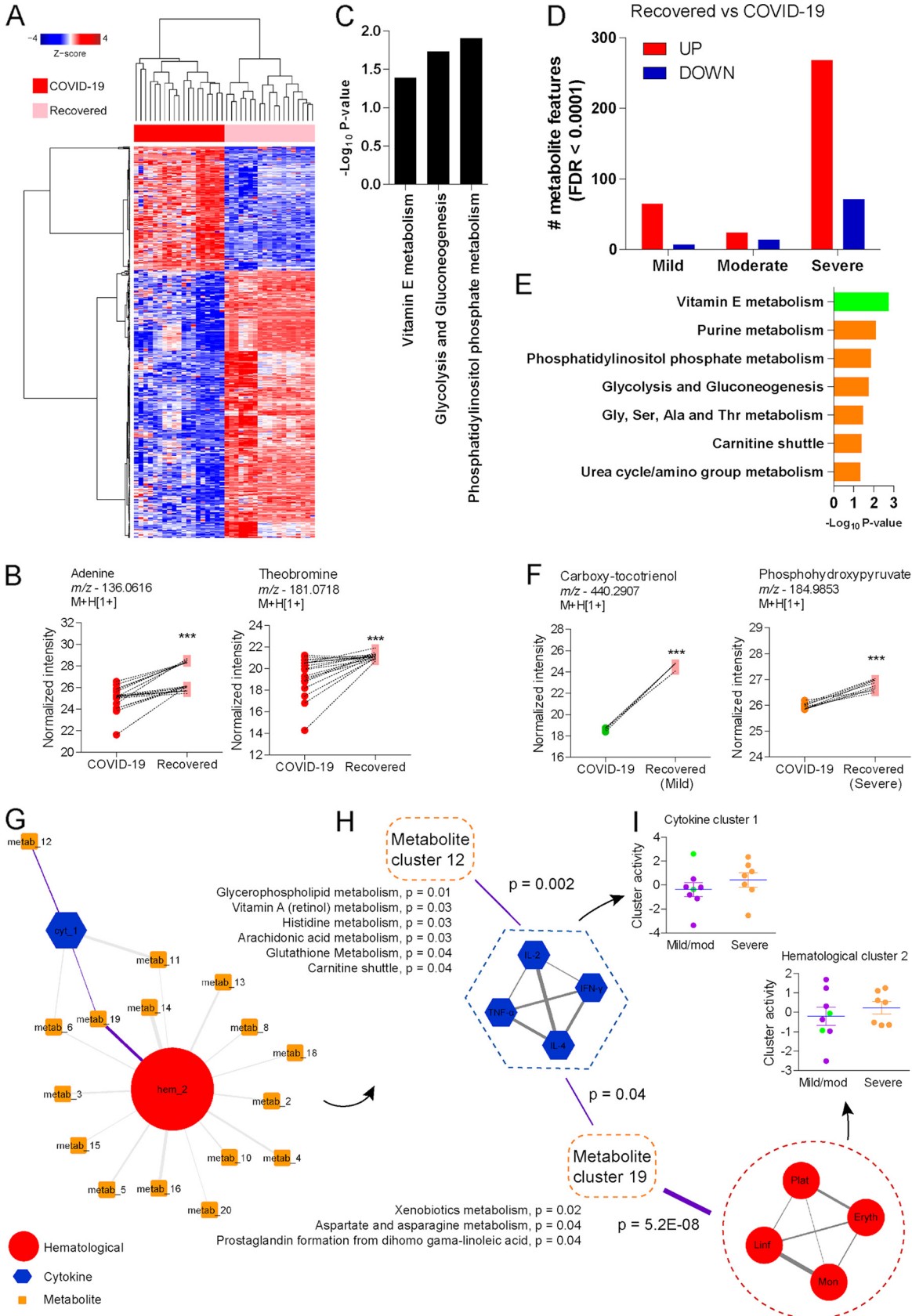

**FIG 5** Metabolomic signatures of recovery from acute COVID-19. (A) Two-way hierarchical clustering of significant metabolite features on a longitudinal follow-up of selected individuals with COVID-19. (B) Differential abundance of adenine and theobromine before and

clinical outcomes of overweight individuals with COVID-19 (37, 38). Other studies reported increased abundance of carnitine and/or acylcarnitines with infection by SARS-CoV-2, and many of them also associate this increase with disease severity (8, 39–42). In contrast, reduced levels of acylcarnitines were also implicated in COVID-19 severity (28, 43, 44). L-Carnitine may reduce the expression of receptors involved in SARS-CoV-2 entry into host cells and inhibit epithelial cell infection (45), and genetic traits inducing higher levels of L-carnitine are associated with reduced susceptibility and severity of COVID-19 (46). Although the regulation and function of carnitine and acylcarnitines during COVID-19 are still under debate, the current evidence points to crucial roles of these metabolites in infection and disease progression.

The infection by SARS-CoV-2 and severity of COVID-19 are associated with the activity of multiple metabolic pathways. However, fatal disease is associated with more perturbations in the plasma metabolome. Pathways such as lysine metabolism and tryptophan metabolism are also associated with levels of oxygen saturation, suggesting a role for amino acids and their derivatives in respiratory distress. In line with these findings, an independent metabolomics study reported altered amino acid catabolism in hypoxic individuals with COVID-19 (47). Hypoxic conditions, such as those observed in individuals with severe COVID-19, can impair mitochondrial function and cellular bioenergetics, causing energy deficits (48). As reported by independent studies, we observed increased abundance of phenylalanine with SARS-CoV-2 infection (32, 49, 50). Although differences between categories of severity did not reach statistical significance, phenylalanine has been implicated as a biomarker of COVID-19 severity (51–53).

The plasma metabolome of individuals with fatal COVID-19 was characterized by the differential activity of a metabolite network hinging on the amino acids glutamate and tryptophan. Many of the metabolites on the network have been previously associated with infection, sex disparities, and disease severity (7, 17, 28, 29, 54). In our study, fatal disease was associated with reduced abundance of oxoproline, which declined over time in an independent cohort of patients who also died due to COVID-19 (16). Oxoproline can be oxidized by oxoprolinase to glutamate, whose abundance was elevated in the plasma of individuals with fatal disease. In addition, levels of glutamyl-glutamate also increased in the plasma of patients who died. These metabolites are involved with glutathione metabolism, the most significant pathway differing between patients who survived or died after severe COVID-19. The balance between glutamate, oxoproline and glutamyl-amino acids can affect the synthesis of glutathione and tissue homeostasis, suggesting that failure in the generation and/or function of glutathione underlies fatal outcomes of COVID-19. Consistent with this hypothesis, SARS-CoV-2 infection impairs the metabolism of glutathione and its antioxidative function *in vitro* (55).

Of interest, our study suggests altered abundance of several lipids in the plasma of individuals with fatal disease, including progesterone, phosphocholine, and lysoPCs. Progesterone diffuses through cell membranes and binds to the progesterone receptor (PR) in the cytoplasm, where PR dimers form a complex that translocates to the nucleus, dampens inflammatory-gene transcription, and promotes expression of growth factors (56). Therefore, progesterone could regulate cellular responses of individuals with COVID-19 (57). At the same time, reproductive steroids may underlie sex disparities in clinical outcomes of COVID-19 (58). Importantly, a randomized, placebo-controlled trial demonstrated enhanced antibody production, reduced production of proinflammatory cytokines, and reduced severity in individuals with COVID-19 who were given estradiol and progesterone early in the course of the disease (59). LysoPCs upregulate adhesion

**FIG 5** Legend (Continued)

after recovery from COVID-19. (C) Mummichog pathway analysis of significant metabolite features. (D) Significant metabolite features before and after recovery from acute COVID-19, stratified by disease severity. (E) Mummichog pathway analysis of significant metabolite features stratified by disease severity. (F) Differential abundance of metabolite features annotated as carboxy-tocotrienol and phosphohydroxypyruvate before and after recovery from acute COVID-19, stratified by disease severity. (G) Integrative hierarchical community network underlying recovery of acute COVID-19. (H) Amplified visualization of nodes linked by purple-highlighted edges on the main network. (I) Cluster activity of cytokine cluster 1 and hematological cluster 2 between patients with mild/moderate and severe disease who recovered from acute COVID-19. ***, $P < 0.001$.

molecules (60) and induce proinflammatory cytokines (61). Reduced levels of these lipids in fatal outcomes could reflect enhanced viral replication, as viruses alter the cell membrane composition during this process (62). In line with our findings, reduced levels of lysoPCs and other phospholipids predict the severity of COVID-19 (9).

Individuals with severe and fatal COVID-19 displayed neutrophilia and lymphopenia, as well as increased levels of CRP, ferritin, and cytokines such as IL-6, IL-10, gamma interferon (IFN-$\gamma$) and TNF-$\alpha$, as described previously (4, 18, 19, 28). The integration of different orthogonal data in our study revealed that metabolite clusters associated with all other data types, but only one connection between hematological and a biochemical cluster reached statistical significance. These results suggest the metabolic response as an integrative hub for the communication between different physiological processes involved in the response to COVID-19. In association with many metabolite clusters, biochemical cluster 2, which includes CRP, D-dimer, and ferritin, predicted the fatality of individuals with COVID-19. Fatal outcome was also predicted by a cluster associated with inflammatory fatty acids (metabolite cluster 21), which emerged as critical molecules in COVID-19 pathogenesis (63, 64).

The understanding of metabolic profiles during convalescence and recovery from acute COVID-19 might reveal mechanisms of protection or even pathology associated with long COVID-19. There are several lines of evidence pointing for a return to baseline levels of metabolites affected by SARS-CoV-2 infection and disease severity, including acylcarnitines and lysoPCs (9, 41). However, metabolite abundance early (1 month) after recovery is still altered (65) and persists 2 to 3 months after nonsevere COVID-19 (41). Other studies evaluating individuals who recovered from more severe phenotypes also found persistent alterations in the plasma metabolome (17, 30, 32). Despite our small sample size, individuals who recovered from acute COVID-19 in our cohort also displayed persistent metabolic alterations. Paired samples from the same individuals revealed that patients with severe COVID-19 are the most affected. Example of altered metabolites include adenine, which inhibits the production of proinflammatory cytokines and bioactive lipids in cellular and animal models of inflammation (66, 67). The pathology of individuals recovering from severe COVID-19 may take longer to resolve, which in turn could stimulate prolonged production of anti-inflammatory metabolites. The metabolism of vitamin E was enriched in plasma of individuals who recovered from mild COVID-19. Vitamin E regulates the immune response and confers resistance to many viral infections (68). Furthermore, vitamin E and derived metabolites, such as tocotrienols, are antioxidant molecules that improve the response to oxidative stress (69) and also regulate inflammation (70). Of note, recovery of all individuals was coupled with higher levels of theobromine in the plasma. This metabolite is found in food and is endogenously metabolized from caffeine. Sickness behavior is characterized by reduced food ingestion, which affects host responses to infections (71), and theobromine levels might indicate improved nutritional status upon recovery or even contribute to the resolution of symptoms such as cough (72). Importantly, data integration revealed that metabolite clusters better explain differences in individuals who recovered from mild/moderate and severe disease.

Overall, the data indicate that metabolomic profiles of individuals with COVID-19 is influenced by a wide range of confounding factors, which were accounted for in our models. However, other factors were not controlled, including individual genetics, SARS-CoV-2 strain, diet, and drug therapy. We were unable to control for treatment, because individuals received different drugs via distinct regimens over the course of disease, including those with mild phenotypes, many of whom self-medicated. Our study is also limited to measurements of few cytokines and laboratorial and hematological parameters that do not reflect the entire complexity of the inflammatory response. Moreover, the small number of individuals followed up after hospital discharge may represent only individualized phenotypes. Importantly, the hypotheses raised by our study need to be validated in independent cohorts and mechanistic studies.

## MATERIALS AND METHODS

**Study population and sample processing.** Individuals with COVID-19 were admitted to the Hospital das Clínicas and Hospital das Clínicas de Campanha or recruited at the Laboratório Prof$^a$ Margarida Dobler Komma at the Federal University of Goiás, Goiânia, Brazil between June 2020 and February 2021, before vaccination rollout. Blood samples were collected in EDTA tubes from 150 individuals who had SARS-CoV-2 infection confirmed by RT-qPCR test from nasopharyngeal swabs or by serological assays to detect specific IgM/IgG antibodies (Eco diagnostics) and from control donors ($n = 27$), who were negative for SARS-CoV-2 infection confirmed by RT-qPCR from nasopharyngeal swabs and serological IgM/IgG tests. Blood samples were collected an average of 13 days from symptom onset (Table S1). Paired blood samples were collected from a subset of 20 individuals with COVID-19 who were followed up after recovery on average of 172 days after symptom onset (Table S2). The criteria defined in the COVID-19 treatment guidelines (National Institutes of Health, USA) and the World Health Organization (73, 74) were used to stratify individuals with COVID-19 into mild disease (individuals presenting various signs and symptoms without shortness of breath, dyspnea, or abnormal chest imaging), moderate disease (individuals presenting radiologically confirmed pneumonitis, hospitalization, and oxygen therapy), severe disease (dyspnea, respiratory frequency of $\geq$30 breaths/min, oxygen saturation [SpO$_2$] of $\leq$93%, and/or lung infiltrates of >50% within 24 to 48 h, including individuals who required monitoring and treatment in an intensive care unit and mechanical ventilation), or fatal COVID-19. Laboratory parameters of liver and kidney function, inflammatory markers and coagulation factors, red blood cells, hemoglobin, platelets, and total and differential leukocytes were determined using automated equipment and a hematology smear, respectively. Blood samples were centrifuged at 1,800 rpm for 10 min to obtain plasma, which was stored at $-80°$C. The research protocol was approved by Ethical Appreciation (CAAE: 30804220.2.0000.5078). All participants provided informed consent according to the regulations of the Human Ethical Committee at the Hospital das Clínicas, Faculdade de Medicina of Universidade Federal de Goiás (UFG-GO). For individuals in the intensive care unit and those who were unable to communicate, consent was obtained from a legally authorized representative (LAR).

**RNA extraction and detection of SARS-CoV-2.** Nasopharyngeal swab samples from individuals with COVID-19 or control donors were processed and analyzed by RT-qPCR. Briefly, RNA extraction was performed with the commercial Qiagen viral RNA minikit (Qiagen, Hilden, Germany), following the manufacturer's instructions. The RNA extracts were subjected to RT-qPCR assay using the Promega Go-Taq Probe one-step RT-qPCR system, according to the manufacturer's protocol. Primers, probe, and synthetic positive control (nCoVPC) were manufactured by IDT (Integrated DNA Technologies, Iowa, USA) and targeted two regions of the N gene (N1 and N2) and the endogenous control, human RNase P gene (RP).

**Cytokine measurements.** Plasma levels of the cytokines IL-2, IL-4, IL-6, IL-10, TNF, and IFN-$\gamma$ were measured in samples using a BD cytometric bead array (CBA) human Th1/Th2 kit (BD Biosciences, San Jose, CA, USA), according to the manufacturer's instructions. Briefly, after sample processing, the cytokine beads were counted using a flow cytometer (FACSCanto II; BD Biosciences, San Diego, CA, USA), and analyses were performed using FCAP Array (3.0) software (BD Biosciences, San Jose, CA, USA). Log$_2$-transformed mean fluorescence intensities were used for statistical analyses and data integration.

**LC-MS/MS.** For metabolomics analyses, cold acetonitrile was added to plasma samples (2:1 [vol/vol]) and subjected to vortex mixing and centrifugation (10 min, 10,000 rpm at 4°C) for protein precipitation. The stable isotopes [$^{13}$C3]caffeine, [$^{15}$N]tyrosine, and progesterone-d9 were used as internal standards, and samples were transferred to injecting vials for LC-MS/MS analysis, which was performed with a high-performance liquid chromatograph (HPLC-UV; 1220 Infinity; Agilent Technologies) coupled with a Q Exactive hybrid quadrupole-Orbitrap high-resolution mass spectrometer (Thermo Fisher). Reverse-phase C$_{18}$ chromatography was performed with Zorbax Eclipse Plus C$_{18}$ columns (4.6 by 150 mm; 3.5 $\mu$m; Agilent) and positive electrospray ionization. All samples were analyzed using a gradient elution program. The binary mobile phases were water–0.5% formic acid with 5 mM ammonium formate (A) and acetonitrile (B). Their gradient elution started with 20% B for 5 min, linearly increased to 100% B in 30 min, and was kept constant for 8 min at 100% B. The eluent was restored to the initial conditions in 4 min to re-equilibrate the column and held for the remaining 8 min. The flow rate was kept at 0.5 mL min$^{-1}$. The injection volume for analysis was 3 $\mu$L, and the column temperature was set at 35°C. The electrospray ionization was operated with the following settings: spray voltage, 3.5 kV; capillary temperature, 269°C; S-lens RF level, 50 V; sheath gas flow rate, 53 L min$^{-1}$; auxiliary gas flow rate, 14 L min$^{-1}$; sweep gas flow rate, 3 L min$^{-1}$. The high-resolution mass spectrometry was carried out in full MS/dd-MS2 mode. The mass range in the full MS scanning experiments was $m/z$ 80 to 1,200. The max IT was set at 200 ms, and AGC target was set at $1 \times 10^6$. For fragmentation acquisition, the top 5 (TopN, 5; loop count, 5) most abundant precursors were sequentially transferred into the C-Trap (AGC target $1 \times 10^5$; max IT 50 ms) for collision. The collision energy for target analytes was 20, 30, and 35 eV. Resolving power was set at 140,000 and 70,000 for full MS and dd-MS2 acquisitions, respectively.

**Bioinformatics and statistical analyses.** Proteowizard software was used to convert .raw files into .mzXML format, and apLCMS R software (75) was used to perform peak deconvolution and detection, to filter noise, to align mass-to-charge ratio ($m/z$) and retention time, and to quantify metabolite features, which are defined by specific $m/z$, retention time, and intensity values for each sample. Pooled human plasma samples were used for quality control (QC) and included in every batch of samples. Replicate samples were summarized based on a Pearson correlation coefficient ($r$) of >0.7. Data were log$_2$ transformed, and features were filtered out by 90% presence in all samples and a coefficient of variation of <0.2 based on QC samples, resulting in 9,893 metabolite features used in further analysis. We used the ComBat function of the sva R package to correct for batch and technical effects. The Mummichog software (version 2) was used to predict activity of metabolic pathways and networks (mass accuracy under

10 ppm) (20). The R package ggplot2 was used to generate volcano, bubble, and Manhattan plots. Heat maps were generated with gplots, and hierarchical clustering was performed with the amap package using Pearson correlation as the distance metric and ward linkage. MZmine v 2.5 software was used to process LC-MS/MS data, with noise set at 1E04 and 1E03 for the MS1 and MS2 levels, respectively. $m/z$ tolerance was set at 0.02 $m/z$ or 10 ppm. ADAP chromatogram builder parameters included the following: minimum group size in number of scans, 5; group intensity threshold, 5E04; and minimum highest intensity, 1E05. Deconvolution parameters included the following: minimum peak height, 1E05; peak duration, 0 to 3 min; and baseline level, 1E04. Isotopic grouping parameters included the following: RT tolerance, 0.25 min; maximum charge, 2; and representative isotope, most intense. Join aligner parameters included the following: weight for $m/z$, 75; RT tolerance, 0.25 min; and RT weight, 25. MS/MS patterns were exported, and fragment similarity searches were performed with METLIN Gen2 (https://metlincloud2.massconsortium.com/) (76) and MyCompoundID (http://www.mycompoundid.org/) (77) (Fig. S7).

Differential abundance was evaluated with moderated $t$ test or moderated $F$ test (analysis of variance [ANOVA]) using the limma R package, which enabled the design of models containing covariables to account for potential confounding factors, including age, sex, and comorbidities. Additional statistics included Tukeýs or Kruskal-Wallis multiple-comparison test. The influence of covariables including age, sex, and comorbidities over the metabolome was evaluated with logistic regression. Multivariable linear regression to identify association between metabolites and $SpO_2$ included age, sex, and comorbidities as covariables. Regressions were performed with glm function of the epicalc R package. FDR was calculated with the Benjamini-Hochberg method.

Data integration was performed using the hierarchical community network approach using python, as described previously (25, 26). Unsupervised hierarchical clustering based on correlation metrics was used to collapse hematological, laboratory, and cytokine data into clusters. For metabolite features, the same hierarchical clustering was applied, but close retention time was enforced within clusters to group different ions reflecting the same metabolite (25). PLS regression was used to calculate associations between clusters, while the significance ($P$ value) of such associations was computed on over 1 million permutations for each pair of data, resampling both samples and features. Each node is a subnetwork, which enables an overview of the leading network and subnetworks for more details at multiple zoom levels. To query the main network, we used t-scores as a primary score in GSEA software. t-scores were obtained from limma output comparing individuals with mild/moderate and severe disease or comparing individuals with severe and fatal COVID-19. To integrate data from recovered individuals, we normalized the data by subtracting values for time point 1 (disease) from those for time point 2 (recovery), whereas biochemical clusters were not included, because many laboratory parameters were unavailable upon recovery. The networks of significantly associated clusters were visualized with Cytoscape v. 3.8.2. Activity of each cluster on a per-sample basis was calculated as follows: sum($z$ score)/square root(number of metabolite features).

**Data availability.** Metabolomics data have been deposited in the Metabolomics Workbench database with the identifier ST002291. The processed feature table used in the analyses is available at Figshare (https://doi.org/10.6084/m9.figshare.22047761.v1). Phenotypic data and annotations are provided in Data Set S1. Code used in the statistical analyses of metabolomics data is provided in Data Set S2.

## SUPPLEMENTAL MATERIAL

Supplemental material is available online only.
**SUPPLEMENTAL FILE 1**, PDF file, 1 MB.
**SUPPLEMENTAL FILE 2**, XLSX file, 0.4 MB.

## ACKNOWLEDGMENTS

This work was funded by Fundação de Amparo à Pesquisa do Estado de Goiás (grant ID 202010267000284) and amfAR (grant ID 110027-67-RGRL) grants to S.G.F. This work was supported by the Serrapilheira Institute (grant number Serra – R-2011-37433) to L.G.G. Graduate students receive scholarships from Coordenação de Aperfeiçoamento de Pessoal de Nível Superior (CAPES). L.G.G., P.R.T.R., B.G.V., and S.G.F. are research fellows with the Conselho Nacional de Desenvolvimento Científico e Tecnológico (CNPq).

We thank all participants of this study. We are thankful to José Clecildo Barreto Bezerra (IPTSP-UFG) for his efforts in setting up a COVID-19 lab at beginning of the pandemic. We are thankful to Hélio Rocha, Nádia do Lago Costa, Ana Carolina Serafim Vilela, Ana Carolina Dourado Leobas, Gabriela Santos Silva, Matheus Henrique Assis de Lima, Luismar Pereira Cardoso, Aline Garcia Kozlowski, Luiz Eterno Xavier, Arthur Christian Garcia da Silva, Luiza Assad Terra, Daniel Fernandes Correia Junior, Cláudia Guimarães, Lorena Ribeiro Alcântara de Sá, Carla Moraes Resende, Larissa Barros Miranda, Tiago Maciel Rego, and João Paulo Scalabrini Brito for their collaboration and technical assistance.

Conceived and supervised the study: L.G.G., S.G.F. Recruited participants, collected samples, epidemiological and clinical data: C.D.P.S., A.R.G.J., A.O.G., M.A.S.B.B., B.G.N.M., S.M. Processed samples: C.D.P.S, D.C.C.A., A.R.G., J.M.M.S., L.C.M., F.P.D.S. Performed and supervised RT-qPCR analysis: D.C.C.D.A., F.S.F., M.S. Performed and supervised cytokine

analysis: C.D.P.S, M.S.F., S.G.F. Performed and supervised LC-MS/MS analysis: G.D.S.L., G.F.D.S., R.R.D.S., G.L.D.A., M.F.R., L.C.D.S., R.C.S., A.R.C., B.G.V. Performed data analysis and generated figures and tables: L.G.G., C.D.P.S. Original draft: L.G.G., C.D.P.S, J.M.M.S., L.C.M., F.P.D.S. S.G.F. Critical discussion, review and editing of the manuscript: L.G.G., C.D.P.S, G.D.S.L., D.C.C.D.A., A.O.G., M.A.S.B.B., F.S.F., M.S., P.R.T.R., M.S.F., V.C., R.C.S., A.R.C., B.G.V., S.G.F. Funding acquisition: L.G.G., R.C.S., A.R.C., B.G.V., M.S., F.S.F., S.G.F. All authors reviewed and approved the manuscript.

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
