## [Reviewer comments · Microbiology Spectrum]

Microbiology Spectrum

Integrated metabolic and inflammatory signatures associated with severity, fatality, and recovery of COVID-19

Luiz Gardinassi, Carolina Servian, Gesiane Lima, Déborah Anjos, Antonio Gomes Junior, Adriana Guilarde, Moara Borges, Gabriel Santos, Brenda Moraes, João Silva, Letícia Masson, Flávia Souza, Rodolfo Silva, Giovanna Araújo, Marcella Rodrigues, Lidya Silva, Sueli Meira, Fabiola Fiaccadori, Menira Souza, Pedro Romão, Mônica Ferreira, Verônica Coelho, Andréa Chaves, Rosineide Simas, Boniek Vaz, and Simone Fonseca

Corresponding Authors: Luiz Gardinassi, Rosineide Simas, Boniek Vaz, and Simone Fonseca, Federal University of Goiás

Review Timeline:

Submission Date:	June 17, 2022
Editorial Decision:	August 12, 2022
Revision Received:	October 18, 2022
Accepted:	February 4, 2023

Editor: Daniel Perez

Reviewer(s): The reviewers have opted to remain anonymous.

Transaction Report:

DOI: <https://doi.org/10.1128/spectrum.02194-22>

August 12, 2022

Dr. Luiz G Gardinassi
Federal University of Goiás
Goiânia
Brazil

Re: Spectrum02194-22 (Integrated metabolic and inflammatory signatures associated with severity, fatality, and recovery of COVID-19)

Dear Dr. Luiz G Gardinassi:

Link Not Available

Sincerely,

Daniel Perez

Journals Department
Reviewer comments:

Reviewer #1 (Comments for the Author):

This study is one of the several examples demonstrating that infection by SARS-CoV-2 induces concerted activity of metabolic and inflammatory responses, that depend on disease severity. Additionally, here it is proposed that these responses can collectively predict clinical outcomes of COVID-19.

The study relies on a relatively large cohort: 150 individuals with mild to severe disease, of which 33 progressed to a fatal outcome. Paired blood samples were collected from a subset of 20 individuals followed up after recovery on average of 172 days after symptom onset.

As far as metabolomics is concerned, high-resolution mass spectrometry was used to investigate small biochemical compounds (< 1500 Da) in plasma. With an untargeted approach, 610 up and 470 downregulated metabolite features (FDR < 0.0001) were

found. It would be extremely useful for the reader to have a list of these compounds as Supplementary Information. Other aspects require a deeper discussion.

- Lines 205-207: why this sort of normalization with respect to the average value in normal subjects has been used only for the comparison between fatal and non fatal disease?

There are several points to be discussed in comparison with existing literature:

- Phe: the levels of this amino acid are increased in COVID-19 patients with respect to controls, but was similar among categories of severity. Up-regulation of this amino acid has been reported by others also by NMR (see for example PLoS Pathog. 2022 18(4):e1010443 and refs. therein), but some of these authors observed a trend in Phe levels which is a function of the disease severity. This information should be mentioned in the discussion.
- Analogously to this study, some of the papers cited above report that metabolome returns to baseline upon recovery from acute COVID-19. In addition to this, here it is reported that the recovery time is different depending on disease severity in the acute phase. But only 20 individuals were followed up after recovery. With a distribution of 4 mild, 6 moderate and 10 severe subjects, I doubt about the statistical significance of this result.
- It is commonly accepted that a certain number of patients develop long-COVID symptoms. Is there any of the 20 follow-up subjects classifiable as suffering from long-COVID?

Reviewer #2 (Comments for the Author):

In this manuscript, Gardinassi et al analyzed the metabolic and immunological alterations associated with COVID-19 of varying severity. They identified multiple metabolites associated with disease severity, as well as varying metabolic trajectories following recovery depending on initial disease severity. Overall, this manuscript addresses an important issue, with recovery processes in particular incompletely understood. However, I have the following major concerns:

1. Statistical analysis of Table S1 data is needed, to determine whether any of those parameters differ between groups
2. Statistical analysis is needed to support findings reported from Figure S1CDE.
3. Control group data is needed for Figure S1D
4. There is insufficient comparison to the extensive prior literature on metabolomics in COVID-19. For example, authors should compare their results to Ghini et al (<https://doi.org/10.1371/journal.ppat.1010443>), which also looked at recovery, though using NMR rather than MS methods. Likewise, at line 157, authors mention that their observation of elevated acylcarnitines differs from the findings of reference 25. While this is correct, the authors' results do match with PMC8686810.
5. Reliance on comparing number of significant features between groups in Figure S2 is unsuitable, since this will be affected by the number of samples in each comparison group, and groups are unbalanced (for example, very different number of participants with and without diabetes).
6. Figure S2 presented data does not match with its description in the text. A comparison of the metabolomes of individuals with COVID-19, classified by sex, as described in the text, should return a single number of significant metabolites that differed between male and female participants. Statistical analysis is also lacking, such as Fisher's exact test to compare the ratio of significant to non-significant features between patient characteristics.
7. Figure 1E seems to indicate that all metabolites affected by SARS-CoV-2 status are correlated with age, sex and comorbidities, and only with one of these parameters. This appears surprising. Is this correct? If not, figure should be adjusted to accurately reflect the data.
8. Lines 176-180: authors should specify the numbers of metabolites that are similar between disease severity levels (matching with carnitine), and the number that differed by severity (matching with N1-acetylpermidine)
9. Figure 2D: the lower pathway activity in the moderate group is surprising, compared to both mild and severe groups. Authors should verify the data, and discuss possible causes in the manuscript.
10. Lines 206-207: please clarify "remaining values are log2 fold change over controls". Are you referring to the data visualization, or to a second set of data transformations applied to the subtracted values?
11. Parameters are needed for LC-MS/MS data processing (eg parameters for deconvolution, noise, RT and m/z tolerance).
12. Support for the annotations discussed in the text should be provided (MS2 spectral matching to library reference or retention time matching to pure standards).
13. Accession number for Metabolights database must be provided prior to publication.

Minor concerns:

1. Code for moderated T-test, moderated F-test and multivariable linear regressions should be provided, either in methods, supplemental, or as a github link (or similar).
2. Line 338/339: It is suggested that the balance between oxoproline and glutamate can be associated with the clinical evolution of COVID-19. Authors should provide a biological hypothesis for this phenomenon, and link this observation to the existing literature.
3. Line 90 typo: "as well" should be "as well as"
4. Line 183: please define or clarify what is intended by "redundant".
5. Table S2 would be more informative if values that deviate from the healthy reference are marked for bloodwork, or alternatively if it's indicated that all values are within normal ranges.
6. Lines 181-182 and 186-189: as described, it's unclear what the difference is between Figure 2D and Figure S3. Please clarify.
7. Line 220: "Figure 4F" should be "Figure 3G"

8. Line 243: "Figure 5C" should be "Figure 4C"
9. Lines 253-257: please clarify what is meant by "cluster activity".
10. Line 318: "abundance of carnitines". Appropriate terminology is "abundance of carnitine and acylcarnitines"
11. Line 321 typo: "are associated the" should be "are associated with the"
12. Line 336 typo: "due COVID-19" should be "due to COVID-19"
13. Line 341 typo: "and binds to"
14. Line 433 typo: "following de manufacturer's" should read "following the manufacturer's"
15. Line 468 typo: ensure that 106 is written as 10⁶ or equivalent formatting.
16. Figure 4A/5G: The yellow highlighting is difficult to see. Switch to a darker color
17. Line 319-320: citation needed for the link between weight and COVID-19 outcomes.

Staff Comments:

Preparing Revision Guidelines

Please return the manuscript within 60 days; if you cannot complete the modification within this time period, please contact me. If you do not wish to modify the manuscript and prefer to submit it to another journal, please notify me of your decision immediately so that the manuscript may be formally withdrawn from consideration by Microbiology Spectrum.

Dear editor, we appreciate the constructive criticism of the reviewers. We have made extensive amendments to the study and below is a point-by-point reply to their comments and concerns.

Reviewer #1 (Comments for the Author):

This study is one of the several examples demonstrating that infection by SARS-CoV-2 induces concerted activity of metabolic and inflammatory responses, that depend on disease severity. Additionally, here it is proposed that these responses can collectively predict clinical outcomes of COVID-19.

The study relies on a relatively large cohort: 150 individuals with mild to severe disease, of which 33 progressed to a fatal outcome. Paired blood samples were collected from a subset of 20 individuals followed up after recovery on average of 172 days after symptom onset.

As far as metabolomics is concerned, high-resolution mass spectrometry was used to investigate small biochemical compounds (< 1500 Da) in plasma. With an untargeted approach, 610 up and 470 downregulated metabolite features (FDR < 0.0001) were found. It would be extremely useful for the reader to have a list of these compounds as Supplementary Information.

We included the supplementary Data S1 which contains phenotypic data and tentative annotations at level 3 of identification according to the Metabolomics Standard Initiative (MSI). We have also annotated several metabolites at level 2 of identification according to MSI, and supporting information is given in Fig. S7 of the revised manuscript.

Other aspects require a deeper discussion.

• Lines 205-207: why this sort of normalization with respect to the average value in normal subjects has been used only for the comparison between fatal and non fatal disease?

In this analysis the goal was to identify significant differences between patients with severe disease that survived or that evolved to a fatal outcome. This additional normalization step here was performed to account for basal abundance of metabolite features in controls to improve the correction of non-biological variation between samples. Moreover, we can readily visualize how much these features also differ from

control samples, meaning that infection indeed affects these metabolites, which also differ between those that survive or die from COVID-19. For example, progesterone differs significantly between severe and fatal samples, but also deviates from control samples, as their mean log₂ fold change over controls is below 0 for both groups. We included traced lines to indicate the control reference in the revised Fig. 3.

There are several points to be discussed in comparison with existing literature:

- *Phe: the levels of this amino acid are increased in COVID-19 patients with respect to controls, but was similar among categories of severity. Up-regulation of this amino acid has been reported by others also by NMR (see for example PLoS Pathog. 2022 18(4):e1010443 and refs. therein), but some of these authors observed a trend in Phe levels which is a function of the disease severity. This information should be mentioned in the discussion.*

We have made extensive amendments and modifications to the discussion of the revised manuscript, comparing our findings to the current literature, which includes phenylalanine and several other metabolites.

Analogously to this study, some of the papers cited above report that metabolome returns to baseline upon recovery from acute COVID-19. In addition to this, here it is reported that the recovery time is different depending on disease severity in the acute phase. But only 20 individuals were followed up after recovery. With a distribution of 4 mild, 6 moderate and 10 severe subjects, I doubt about the statistical significance of this result.

The metabolic activity of recovered individuals is an important issue that is still under debate. During the revision, we re-analyzed all data to confirm the previous results and found inconsistencies that are now corrected. The comparison between recovered individuals with controls revealed that the metabolic profile of recovered individuals did not return to baseline at the time of sample collection. This was corrected in revised manuscript. Because of the sample size, we decided not to subclassify them when comparing to controls. These results are shown in the revised Fig. S5, which are in line with most studies evaluating patients that recover from COVID-19. Most of them report persistent alterations, even in individuals recovered from mild disease. We did subclassify the recovered individuals when analyzing the paired samples (those from the same individuals during disease and after recovery). We understand our sample size for

recovered individuals is small and this was pointed out as a limitation of our study in the discussion. However, as the scientific community moves towards precision medicine, small sample sizes will be a frequent challenge. One way to overcome it is by integrating different orthogonal data as we performed in this study. We included additional analyses showing metabolite clusters that differ between individuals that recovered from severe disease compared to mild/moderate disease in Fig. S6 and described it in page 12, lines 284 - 288. We have also discussed these issues in more details at pages 15-16, lines 362 - 384 of the revised manuscript.

It is commonly accepted that a certain number of patients develop long-COVID symptoms. Is there any of the 20 follow-up subjects classifiable as suffering from long-COVID?

As described in Table S2, many of the individuals recovered from acute COVID-19 reported symptoms that could be related to long-COVID-19. However, we did not explore this issue here, as we believe other parameters are needed to ascertain this phenotype and exclude those with symptoms due other factors (ex: other infection, comorbidity, among others).

Reviewer #2 (Comments for the Author):

In this manuscript, Gardinassi et al analyzed the metabolic and immunological alterations associated with COVID-19 of varying severity. They identified multiple metabolites associated with disease severity, as well as varying metabolic trajectories following recovery depending on initial disease severity. Overall, this manuscript addresses an important issue, with recovery processes in particular incompletely understood. However, I have the following major concerns:

1. Statistical analysis of Table S1 data is needed, to determine whether any of those parameters differ between groups

We included results from statistical analyses in the revised version of Table S1.

2. Statistical analysis is needed to support findings reported from Figure S1CDE.

We included results from statistical analyses in the revised version of the Figure S1.

3. Control group data is needed for Figure S1D

We included data from control group for most biochemical parameters. However, we were unable to measure coagulation functions in the same samples. This observation is written in the legend of the revised Fig. S1 and we emphasize that these data are not critical to the conclusions of our study.

4. There is insufficient comparison to the extensive prior literature on metabolomics in COVID-19. For example, authors should compare their results to Ghini et al (<https://doi.org/10.1371/journal.ppat.1010443>), which also looked at recovery, though using NMR rather than MS methods. Likewise, at line 157, authors mention that their observation of elevated acylcarnitines differs from the findings of reference 25. While this is correct, the authors' results do match with PMC8686810.

We have made extensive amendments and modifications to the discussion of the revised manuscript, comparing our findings to the current literature, which includes acylcarnitines and several other metabolites.

5. Reliance on comparing number of significant features between groups in Figure S2 is unsuitable, since this will be affected by the number of samples in each comparison group, and groups are unbalanced (for example, very different number of participants with and without diabetes).

We performed different analyses to evaluate to which extend confounders influence the plasma metabolome in our cohort. The results are described in page 7, lines 154 – 171 and shown in Figures 1E, 1F and Figure S2 of the revised manuscript.

6. Figure S2 presented data does not match with its description in the text. A comparison of the metabolomes of individuals with COVID-19, classified by sex, as described in the text, should return a single number of significant metabolites that differed between male and female participants. Statistical analysis is also lacking, such as Fisher's exact test to compare the ratio of significant to non-significant features between patient characteristics.

We performed different sets of statistical analyses to evaluate confounding variables. First, we performed logistic regression to identify association of covariables with metabolite features, which demonstrates positive and negative associations with each

tested covariable, as shown in revised Figure 1E. We then compared the overlap of significant metabolite features associated with infection and those associated with covariables, as shown in the revised Figure S2. To confirm this influence, we then performed logistic regression to understand how much each covariable influences the metabolome-wide association with infection, by comparing the effect over the measurement of association (regression coefficient) using models with or without the covariable of interest. We found that all covariables affected the coefficients to some extent (revised Figure 1F). Sample size could influence these analyses, but it would only affect the precision of the estimates. Our goal here was not to precisely identify the specific features affected by each confounder, but rather understand whether they needed to be accounted for in further statistical models.

7. Figure 1E seems to indicate that all metabolites affected by SARS-CoV-2 status are correlated with age, sex and comorbidities, and only with one of these parameters. This appears surprising. Is this correct? If not, figure should be adjusted to accurately reflect the data.

The previous circo plot indicated the number of significant features overlapping between SARS-CoV-2 infection and covariables. However, we performed extensive modifications of the analyses of confounders and their visualization to accurately reflect the results.

8. Lines 176-180: authors should specify the numbers of metabolites that are similar between disease severity levels (matching with carnitine), and the number that differed by severity (matching with N1-acetylspermidine).

The analyses shown in Figure S3 compares only groups of COVID-19, which was incorrectly described in the previous version of the manuscript. Because of the significant effect of infection, we removed the control donors from this analysis to evaluate metabolites that differ by severity of COVID-19. We improved the description in page 8 lines 189 - 193.

9. Figure 2D: the lower pathway activity in the moderate group is surprising, compared to both mild and severe groups. Authors should verify the data, and discuss possible causes in the manuscript.

During the revision, we re-analyzed all data to confirm the previous results and found inconsistencies that are now corrected. Pathway activity of all groups are now consistent with the reported number of differentially abundant metabolite features (DAM) when compared to control samples. There was a dramatic reduction in the number of pathways enriched for the severe group, but the other groups remained mostly the same. Importantly, these changes do not affect the conclusions of our study.

10. Lines 206-207: please clarify "remaining values are log2 fold change over controls". Are you referring to the data visualization, or to a second set of data transformations applied to the subtracted values?

This refers to the normalization performed by subtracting the mean log₂ intensity values of control donors, which results in the log₂ fold change in metabolite abundance of each COVID-19 samples compared to controls. We have modified the text to clarify this issue in page 9, lines 205 – 208 of the revised manuscript.

11. Parameters are needed for LC-MS/MS data processing (eg parameters for deconvolution, noise, RT and m/z tolerance).

The parameters for processing LC-MS/MS data are described in the material and methods, page 20, lines 479 – 485 of the revised manuscript.

12. Support for the annotations discussed in the text should be provided (MS2 spectral matching to library reference or retention time matching to pure standards).

We added Fig. S7 showing support for annotation of metabolites matching MS₂ spectra in reference libraries.

13. Accession number for Metabolights database must be provided prior to publication.

Metabolights was taking too long to review and release data. Therefore, we chose to change the database for Metabolomics Workbench, whose accession number is now provided in the text and data is scheduled to be released on 2022/10/19.

Minor concerns:

1. Code for moderated T-test, moderated F-test and multivariable linear regressions should be provided, either in methods, supplemental, or as a github link (or similar).

Code used for the statistical analysis, including moderated T-test, moderated F- test and multivariate linear and logistic regressions is now given in the supplementary material. I was not able to upload R files (.R) or files over 20MB (processed, transformed, filtered, and normalized feature table) to the supplementary material. Therefore, the processed feature table can be downloaded from the Metabolomics Workbench database and the code is shown in Data S2.

2. Line 338/339: It is suggested that the balance between oxoprolin and glutamate can be associated with the clinical evolution of COVID-19. Authors should provide a biological hypothesis for this phenomenon, and link this observation to the existing literature.

We have evaluated this phenomenon in more details and we believe this balance is related to glutathione metabolism. We have discussed this issue and linked to existing literature in page 14, lines 329 – 3338.

3. Line 90 typo: "as well" should be "as well as"

It was corrected.

4. Line 183: please define or clarify what is intended by "redundant".

We modified the text to better reflect the current results.

5. Table S2 would be more informative if values that deviate from the healthy reference are marked for bloodwork, or alternatively if it's indicated that all values are within normal ranges.

All the values returned to normal ranges, and this indication has been included in the Table S2 of the revised manuscript.

6. Lines 181-182 and 186-189: as described, it's unclear what the difference is between Figure 2D and Figure S3. Please clarify.

We have modified the text accordingly to clarify that Figure S3 represents differential abundance analyzed with moderated F-test only between groups of individuals with COVID-19.

7. Line 220: *"Figure 4F" should be "Figure 3G"*

It was corrected.

8. Line 243: *"Figure 5C" should be "Figure 4C"*

It was corrected.

9. Lines 253-257: *please clarify what is meant by "cluster activity"*.

Cluster activity refers to the collapsed values of clusters. In other words, it is a score reflecting the summarized abundance of its members. It was taken by the formula: activity = sum (z-score) / square root (number of metabolite features). This was included in the material and methods at page 21, lines 511 – 512.

10. Line 318: *"abundance of carnitines". Appropriate terminology is "abundance of carnitine and acylcarnitines"*

It was corrected.

11. Line 321 typo: *"are associated the" should be "are associated with the"*

It was corrected.

12. Line 336 typo: *"due COVID-19" should be "due to COVID-19"*

It was corrected.

13. Line 341 typo: *"and binds to"*

It was corrected.

14. Line 433 typo: *"following de manufacturer's" should read "following the manufacturer's"*

It was corrected.

15. Line 468 typo: *ensure that 106 is written as 10⁶ or equivalent formatting.*

The formatting was corrected for this and other similar typos.

16. Figure 4A/5G: The yellow highlighting is difficult to see. Switch to a darker color

We modified it with a darker color.

17. Line 319-320: citation needed for the link between weight and COVID-19 outcomes.

We included references describing the link between obesity and COVID-19.

February 4, 2023

Dr. Luiz G Gardinassi
Federal University of Goiás
Goiânia
Brazil

Re: Spectrum02194-22R1 (Integrated metabolic and inflammatory signatures associated with severity, fatality, and recovery of COVID-19)

Dear Dr. Gardinassi:

I apologize for the long delay in the handling of your manuscript. It has been a bit rough the last couple of months. Your manuscript has been accepted, and I am forwarding it to the ASM Journals Department for publication. You will be notified when your proofs are ready to be viewed.

Sincerely,

Daniel Perez
Editor, Microbiology Spectrum
